# Determination of "Neutral"–"Pain", "Neutral"–"Pleasure", and "Pleasure"–"Pain" Affective State Distances by Using AI Image Analysis of Facial Expressions

Hermann Prossinger [1,*], Tomáš Hladký [2], Silvia Boschetti [2,3], Daniel Říha [2,3] and Jakub Binter [2,3]

[1] Department of Evolutionary Biology, Faculty of Life Sciences, University of Vienna, A-1030 Vienna, Austria
[2] Faculty of Humanities, Charles University, 182 00 Prague, Czech Republic; tomas.hladky@fhs.cuni.cz (T.H.); silvia.boschetti@natur.cuni.cz (S.B.); daniel.riha@fhs.cuni.cz (D.Ř.); jakub.binter@fhs.cuni.cz (J.B.)
[3] Faculty of Science, Charles University, 128 00 Prague, Czech Republic
* Correspondence: hermann.prossinger@univie.ac.at

**Abstract:** (1) Background: In addition to verbalizations, facial expressions advertise one's affective state. There is an ongoing debate concerning the communicative value of the facial expressions of pain and of pleasure, and to what extent humans can distinguish between these. We introduce a novel method of analysis by replacing human ratings with outputs from image analysis software. (2) Methods: We use image analysis software to extract feature vectors of the facial expressions neutral, pain, and pleasure displayed by 20 actresses. We dimension-reduced these feature vectors, used singular value decomposition to eliminate noise, and then used hierarchical agglomerative clustering to detect patterns. (3) Results: The vector norms for pain–pleasure were rarely less than the distances pain–neutral and pleasure–neutral. The pain–pleasure distances were Weibull-distributed and noise contributed 10% to the signal. The noise-free distances clustered in four clusters and two isolates. (4) Conclusions: AI methods of image recognition are superior to human abilities in distinguishing between facial expressions of pain and pleasure. Statistical methods and hierarchical clustering offer possible explanations as to why humans fail. The reliability of commercial software, which attempts to identify facial expressions of affective states, can be improved by using the results of our analyses.

**Keywords:** image processing; artificial intelligence; facial expressions; affective state expression; facial pain expression; facial pleasure expression; BDSM videos; hierarchical agglomerative clustering; autoencoder neural network

## 1. Introduction

This manuscript reports results that are extensions of the pilot study published in 2021 [1]. There, we investigated, using AI, whether it is possible to distinguish the facial expression of pain from that of pleasure in women.

Consider the following scenario: two persons are having a face-to-face conversation (rather than telephoning). In addition to the exchange of words, the conversants unavoidably register each other's facial expressions. Whether these facial expressions convey and, if yes, how much information is the starting point of the research presented in this paper.

The information transmitted is the superposition of three contributions: acoustic, facial, and semantic. Anyone who attempts to decompose this superposition is faced with extraordinary challenges. A promising approach would be to devise a laboratory or field work setting wherein actors convey their affective states of pain and of pleasure (hopefully convincingly) to an audience. By turning off the microphone or erasing the soundtrack, we are in the position to analyze what is being solely transmitted via facial expressions.

Aim: we wish to show how the use of AI technology that analyzes images is superior to humans' abilities to solve these same facial expression decoding challenges.

### 1.1. Overall Benefits of the Insights We Present in This Manuscript

Due to human evolution, the ability to produce facial expressions does not infer the ability to perceive them. In fact, the shortcoming in this lack of inference is the perception part. We can gain insights by overcoming this shortcoming by relying on technology as an alternative. The result(s) obtained by image analysis software does not, as we will show, reproduce the human result. It seems that computer vision is not compromised by a human brain's visual perception limits. Therefore, we argue that the insights gained are beneficial to our understanding and appreciation of visual perception mechanisms.

### 1.2. Using AI as a Novel Approach to Analyzing Facial Expressions of Pain and Pleasure

One way to perform this analysis would be to query the members of an audience as to their evaluation of presented facial expressions. Another approach is to analyze the expressions using image analysis software. We present research results of this latter approach in this paper.

There exists a considerable amount of literature [2–6] dealing with the recognition and identification of affective states putatively communicated by facial expressions. Novel in the approach we present here are the following: (1) to use one AI tool to extract faces from video frames; (2) to use another AI tool (an autoencoder) to determine their dimension-reduced feature vectors; (3) to quantify their differences (if any); and (4) to implement the machinery of multivariate statistical analysis, combined with clustering algorithms, to detect noise contributions and identify patterns.

For technical and historical reasons, we limit ourselves, in this paper, to focusing on actresses while they express three affective states: pain, pleasure, and neutral.

### 1.3. Previous Reasearch into Facial Expression of (Intense) Affective States

While interpersonal interaction via verbal exchanges involves language, facial expressions are also, arguably, a further method to "advertise" one's affective state and even more so during many social interactions in which the acoustic channel is blocked (such as a pantomime or an interaction through a window or while observing from a distance).

For one person to estimate the affective state of some other person, his/her decoding ability is the foremost prerequisite. Further support for the hypothesis of universality comes from intercultural studies that have documented that expressions of emotion, among them joy, fear, sadness, anger, and surprise (but perhaps neither pain nor pleasure; both are not emotions), are to a large degree universally understood [7–9]; this is also supported by methodologies using AI [10,11].

Many expressions of affective states are closely linked to these in combination; indeed, there is a widespread consensus in psychology that there exist only a small number of emotions expressed and experienced solely. Some of these are also characterized by specific—and identifiable—facial muscle contractions [9].

This inferred existence of the in-between ones has led to a codification called FACS (Facial Action Coding System; [9]). It relies on so-called action units (AU); each unit is based on the contraction of a subset of facial muscles to create, in concerto, the intended facial expression. The derived methods are used both descriptively (in the analysis of the behavior of an individual) and also prescriptively as a template for the animation of facial expressions [12]. A study [13] of several animations that have been used in computer graphics focused on the distinction between the facial contribution and the perceived affective states associated with pain and with sexual climax (pleasure). That study succeeded in demonstrating a difference of mental representations within the onlooker of the two affective state expressions.

One can, of course, attempt to down-regulate the expressiveness and thereby control the amount of information shared (a poker face comes to mind) but there may be limits in the event of extreme experiences. In the context of sexuality, this issue of pain resembling grimace during sexual climax has already been described in the so called "Kinsey Report" (1953). Further examples involving other social situations have recently come to the fore [14].

Pain and pleasure, as signals of extremely important experiences in the lives of humans, should be sufficiently well-recognizable by social animals, which humans certainly are. There is an ongoing debate about the communicative value of these facial expressions [15]. One of the arguments is that they are essential because they provide the interacting partner with a signaling value that he/she can attribute correctly [16]. One of the counterarguments is that the groups of muscles contracted during the intense affective states of pain and pleasure are similar and therefore the only inferable information relates to intensity but does not provide any valence of the signal [13,17].

Thus, there is a possibility that, even if a difference is indeed present, human perception is not calibrated well enough to reliably and reproducibly distinguish such details. Perhaps human vision is unable to adequately cope with the noisiness of the signal and humans may perceive it as individual differences in expression. In essence, the research involving facial expressions perceives a limitation in how to overcome these difficulties in traditional field studies, arguably a gap in available methodology [6]. AI facial analysis should be able to identify such differences, thus bridging this gap and implying that it is superior to human vision in this regard. Already in our pilot study [1], this distinguishability was found in women but not in men. Based on these published outcomes, we increased the sample size but restricted the study to actresses only.

Facial expressions in some cases seem to be ambiguous to raters whenever they are not presented with further cues (such as vocalization or body posture); stimuli are oftentimes acted instead of being real. In one study [18], actors had been trained to act out fear and anger. The accuracy of their acting ability for the affective state of fear was tested by comparisons with real-life recordings; the acted expressions differed greatly, it was found, from the real-life recordings and were more difficult to recognize. In our attempt to further increase the demands of rigor regarding facial expressions of pain and of pleasure, we used real bondage, discipline, and sadomasochism (BDSM) acts performed by professional actors in commercially available videos.

The use of BDSM videos of private persons would infringe on their privacy rights and we could not reliably infer which affective state is being expressed (we would have to rely on their statements). In the case of BDSM videos with professional actresses, on the other hand, we can be certain that the displayed affective state is expressed (being monitored by the director during production) while the issue of privacy rights is moot. Even though the actresses are professionals and are probably anticipatory of what the consumer is seeking or expecting, the stimuli should be considered semi-naturalistic since we, the authors, intentionally chose companies and brands that have a reputation for their realism. There are indicators (such as bruises) that the scenario is not enacted symbolically but that the actresses' experiences are genuine.

### 1.4. Novelty of the Approach Presented in This Paper

While the analyses of facial expressions of pain and pleasure by BDSM actresses are, we claim, novel, per se, we go several steps further. We rated the facial expressions using image analysis software and other AI tools. We present outcomes that indicate to what extent image analysis software is more reliable than human ratings of the facial expressions of pain and pleasure.

We extracted feature vectors from the images and quantified differences between the facial expressions by calculating the Euclidean distance between the dimension-reduced feature vectors (thus defining the affective state distances). We used these distances to determine their maximum likelihood (ML) distributions, quantified the noise, and looked for patterns of these noise-free dimension-reduced feature vectors, which imply patterns in the actresses' facial expressions.

### 1.5. Fields of Study in Which the Results Are of Importance

Applications using image analysis software have become standard in the world of AI. However, because the facial expression of pleasure, for instance, is unreliable, there is a

potential confusion with the facial expression of pain. The results we present here quantify the putative unreliability, which should contribute to the development of more refined algorithms in image analysis software attempting to distinguish these affective states.

## 2. Materials and Methods

### 2.1. Materials

We scanned 20 BDSM videos, each showing professional actresses in action. In each video chosen, using the development of the plot as a reference, five frames with almost format-filling faces—three frames displaying neutrality, one frame pain, and one frame pleasure—were selected. Our database thus consists of 100 images displaying three facial expressions by twenty actresses (no one actress was in more than one video). Ages of the actresses were not revealed in the (sales) texts describing the videos but they could be estimated via their facial features and body attributes; the actresses appeared to be within the 20 to 35-year age bracket.

### 2.2. Methods

We used AI image analysis software to: (a) extract a rectangle in each of the 100 frames containing the faces, plus negligible borders; (b) apply a suite of image analysis routines in the software package (MATHEMATICA® v12.4 from Wolfram Technology) to align the five faces for each actress in each video; (c) implement feature extraction algorithms to construct a feature vector for each face; (d) dimension-reduce the five feature vectors to five 2D vectors (which we call dimension-reduced feature vectors) for each actress's five facial expressions; and (e) calculate the Euclidean distances between pairs ("neutral"–"neutral", "neutral"–"pain", "neutral"–"pleasure", and "pleasure"–"pain") of these dimension-reduced feature vectors. Details and code of the software implementation are listed in Appendix A.

We consider the pain–pleasure dimension-reduced feature vector distances to have been drawn from a statistical population with an unknown parametric distribution. In this paper, we restricted ourselves to four parametric distributions: normal, log-normal, Gamma, and Weibull. We estimated the ML distribution [19] of the pain–pleasure distances and calculated both the mode and expectation value, as well as the highest density interval at 95% uncertainty (HDI$_{95\%}$) [20].

We attempted to quantify noise and looked for patterns as well as possible pattern structures after noise elimination. To do so, we generated a $20 \times 5$ matrix (in which the first three columns are the three distances of neutral from the neutral mean; the fourth column is the distance of pain from the neutral mean; and the last column is the distance of pleasure from the neutral mean). We scaled each column by its mean and centered it by subtracting a vector of 1s. We then performed a singular value decomposition (SVD) [21,22] to determine how many singular values contribute to the signal. We found that the sum of the first three singular values squared extracted 89.8% of the square of the Frobenius norm [21] of the matrix (which we call the scaled, shifted distance matrix). We then applied an unsupervised clustering algorithm (agglomeration cluster algorithm) to determine whether the rendition of affective states by the actresses are segmented into clusters.

## 3. Results

For each actress, the feature vector of each of the five frames was extracted using image analysis software. These five feature vectors for each actress were then dimension-reduced to 2D using an autoencoder (a neural network [22–24] with—in our case—seven layers; Figure 1). We also determined the 2D coordinates of the mean of the three neutral displays for each actress (because we argue that they have a commonality, although not necessarily equality). A result for one actress is shown in Figure 2.

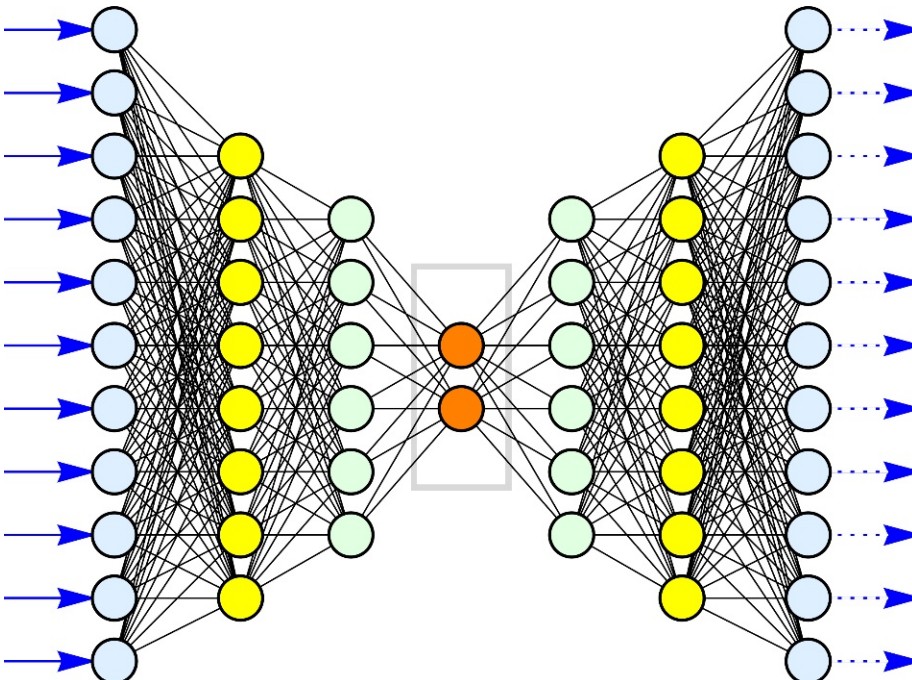

**Figure 1.** A symbolic rendition of an autoencoder used for dimension reduction. The 11 inputs (in the drawing) are represented by blue arrows from left to right. The inputs are conventionally labelled neurons (hence the name "neural network"). Each input neuron (light blue) has as many outputs as there are neurons in the next layer. Each 'yellow' neuron thus has 11 inputs/edges (represented as thin black lines). Each neuron in the yellow layer has as many outputs as there are neurons in the next layer: six outputs for each 'yellow' neuron and therefore eight inputs for each 'green' neuron of the next layer. It continues: each 'green' neuron has as many outputs as there are neurons in the next layer (consisting of two 'orange' neurons) and each 'orange' neuron has six inputs. The light blue neurons on the left are called the input layer while the light blue neurons on the right are called the output layer. The yellow layers, the green layers, and the orange layer are called the hidden layers. An autoencoder always has the same number of output neurons as it has input neurons. The number of hidden layers is part of the design by the engineer constructing the autoencoder. The numerical values along the black edges between neurons are determined by an algorithm. The autoencoder attempts to produce an output equal to the input (hence the name 'autoencoder') without being an identity mapping. An important feature for modern autoencoders is the ability to cut (set to zero) certain interconnections or make them numerically very small (usually by using a sigmoid function). The central layer is called the code. If the inputs are the feature vectors, then the numerical values of the code are the components of the dimension-reduced feature vector. If this is a successful autoencoder, it detects nonlinear combinations between the components of the (input) feature vector that can be represented by two variables.

We found that the pain–pleasure distances were rarely less than the pain–mean neutral and/or pleasure–mean neutral distances. In fact, the 2D points (mean neutral, pain, and pleasure) for most actresses often formed a near-isosceles triangle (occasionally one close to being equilateral).

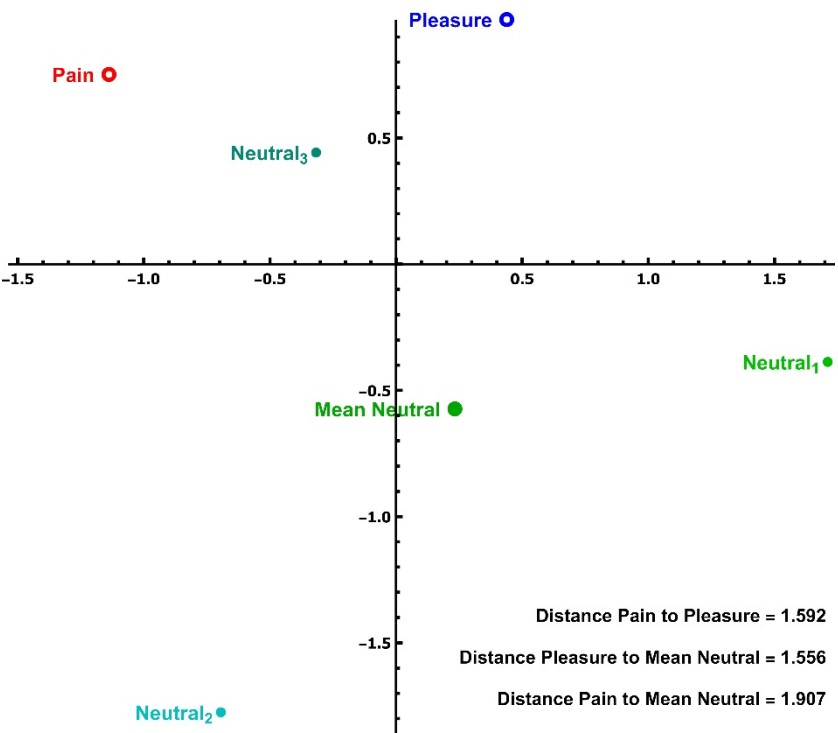

**Figure 2.** The locations of the dimension-reduced feature vectors of the facial display of the (labeled) affective states of one actress. In addition to these five points, we rendered (and used in subsequent calculations) the arithmetic mean (center of mass) of the neutral states, which we call 'mean neutral' for this female. Distances of the dimension-reduced feature vectors for this female are also displayed.

The distances between pain and pleasure were Weibull-distributed. The parameters of the ML distribution, its mode, expectation, and HDI$_{95\%}$ [20] uncertainty interval are listed in Table 1 and the pdf of this ML distribution is displayed in Figure 3.

**Table 1.** Parameters of the ML distribution of the pain–pleasure distances of the 20 actresses; it is a Weibull distribution. E is the expectation and HDI$_{95\%}$ is the 95% highest density interval [20].

| Parameters | ML Numerical Values |
|---|---|
| WeibullDistribution$[k, \lambda]$ | $k = 4.34 \; \lambda = 2.25$ |
| Mode | 2.12 |
| E | 2.05 |
| HDI$_{95\%}$ | $(s_1, s_2) = (0.99, \; 3.06)$ |

The sum of the first three singular values squared of the scaled, shifted distance matrix explains 89.8% of the square of the Frobenius norm of this matrix. We therefore looked for possible clusters in this 'smoothed' (noise-free) pattern matrix. We used the hierarchical agglomerative clustering algorithm and detected six clusters, of which the last two are isolates. Details of the clusters are described in the figure caption of Figure 4.

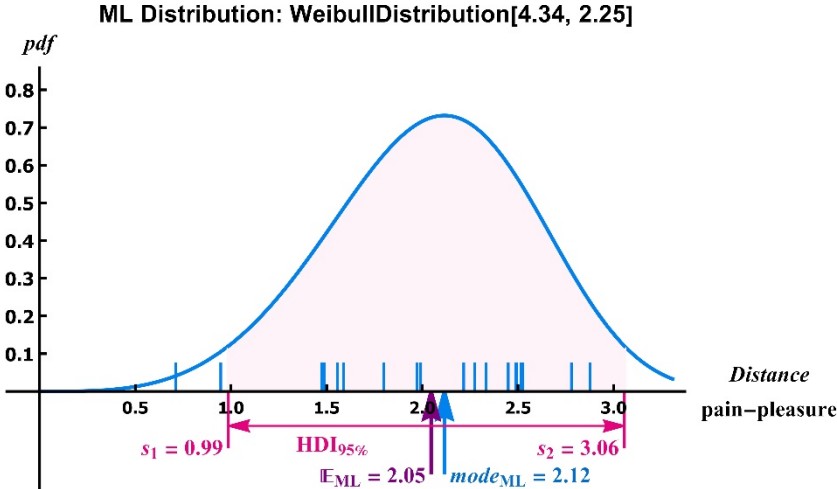

**Figure 3.** The ML distribution of the pain–pleasure distances for all 20 actresses. The distribution is asymmetric, thus displaying the interval mean $\pm$ SD is not meaningful. We used the $\text{HD}_{95\%}$ interval [20] to display the uncertainty. The ML mode and ML expectation are close to midway between the ends of the $\text{HDI}_{95\%}$ interval, and, furthermore, the ends of this interval are very close to $\frac{1}{2}\times$ and $\frac{3}{2}\times$ the mode. The shaded area under the pdf-curve has an area of 95%.

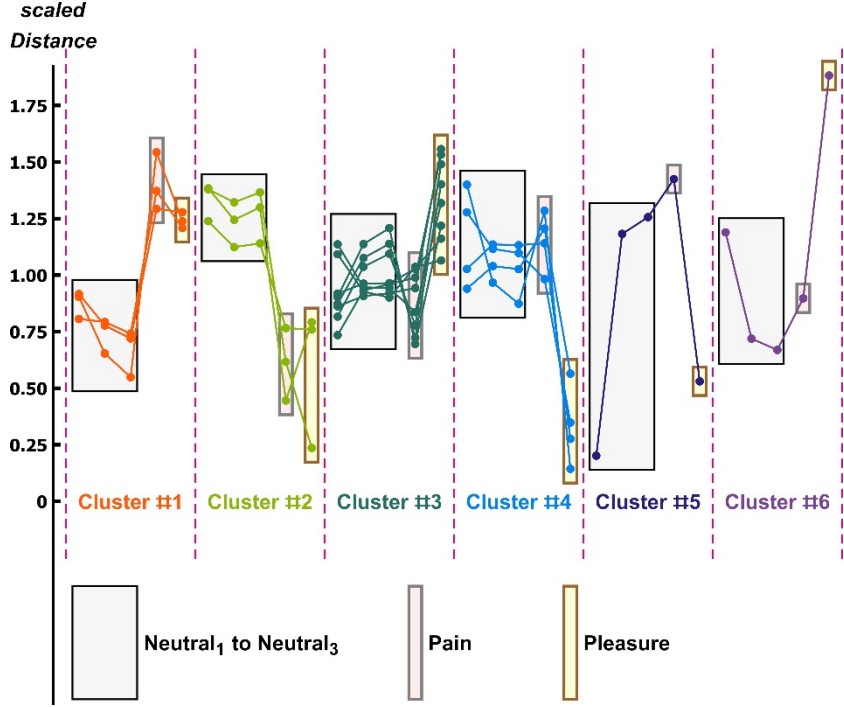

**Figure 4.** The clusters detected in the scaled, shifted pattern matrix. Distances are scaled by the neutral mean. There are six clusters with membership sizes: 3, 3, 8, and 4, along with two further clusters containing singletons (isolates). We observed the following: (i) In Cluster #1, the distances of pain and pleasure from the neutral mean are nearly equal and larger than those of the neutrals to the neutral mean. (ii) In Cluster #2, the distances of pain and pleasure to the neutral mean are also nearly equal but considerably less than the distances of the neutrals to the neutral mean. (iii) In Cluster #3, there is hardly any overlap between the pain and pleasure distances, and the pain distances overlap the neutral distances to the neutral mean. Additionally, the pleasure distances are larger than the pain distances. (iv) In Cluster #4, the pain distances overlap the neutral distances to the neutral mean, but the pleasure distances do not; the latter are, furthermore, much smaller than the other distances.

## 4. Discussion

The affective state distances between the dimension-reduced feature vectors of the expressions of pain and of pleasure are not small; they are, however, comparable with the distances between the three facial expressions of neutral. The facial expressions of pain and of pleasure are distinct for the image analysis software. We contrast this result with the observation that human raters cannot successfully distinguish between these facial expressions [15,25,26].

The three neutral faces of each actress are consistent within Cluster #1 and Cluster #2 but are inconsistent in Cluster #3 and Cluster #4, as well as inconsistent overall (Figure 4). After reviewing (by visual inspection) the frames used for the analysis, we noticed that the neutrality appeared to depend on the camera setting. This effect, which we could not rigorously quantify in this paper, is not a limitation. Raters of the actresses were also dependent, during their attempted identification of the facial expression of pain and pleasure, on the lighting and camera setting. Distances between the dimension-reduced feature vectors of each display of neutral are sometimes comparable to the distances of the pain–neutral mean or pleasure–neutral mean (Figure 4).

The affective state distances between the display of pain and the neutral mean varied considerably. Part of this variation, we discovered, was due to noise. Removal of noise via SVD resulted in distances that were members of one of four clusters (except for two isolates). It is remarkable, we argue, that the distributions of noise-free distances not only cluster but cluster into four (non-trivial) clusters. We suspect that the clustering is an effect ascribable not only to the camera setting but also to the commonalities of expression among the actresses in each cluster.

The human visual system (we discovered) is not well-equipped to rate the differences between the facial expressions of pain and pleasure, whereas the AI methods we present here are (as distances between the dimension-reduced feature vectors). We do note an added superiority of the AI image analysis, SVD, and hierarchical agglomerative clustering: we can quantify the noise and remove it prior to the identification of clusters. We do not argue that training removes errors. A recent publication [26] demonstrated that training (and expertise) does not remove all perception errors by humans.

Our analyses include a word of caution to researchers investigating the postulated indistinguishability of facial expressions of pain and of pleasure, especially if they are relying on data obtained via fieldwork, such as when evaluating ratings. The analyses presented here deals with how well image analysis software of facial expressions makes the distinction between the expression of pain versus that of pleasure possible. As we have documented, there is a small but not-to-be-neglected fraction of noise in the dataset. We must infer that this (statistical) noise is also responsible for making it more difficult for humans to correctly distinguish between these two affective states.

By implication, we conclude that our image analysis software's ability to distinguish between expressions of pain and of pleasure is superior to the human ability to do so.

## 5. Conclusions

We have discovered that image analysis software can be used to construct feature vectors of the facial expressions of pain and of pleasure, and compared them with the feature vectors of the neutral facial expressions.

Image analysis software of affective states has at least one helpful and one ominous use. The ominous one is unwanted supervision, which can be used to monitor the affective states of unsuspecting 'victims' (as in, for example, police states). The (numerous) helpful ones are those that allow for the monitoring of dangerous situations, such as distress, accident consequences, bodily harm, or situations of enforced, unwanted compliance (such as in violent situations). We repeat: properly developed image analysis software is needed to reliably distinguish between facial expressions of pleasure and pain.

The analyses of the outcomes provided considerable information. First, we found that AI methods can more reliably distinguish between pain and pleasure than humans

can (thus succeeding in achieving our aim). Secondly, the reduced feature vector distances (the affective state distances) between pain and pleasure may be comparable to the pain to neutral distances, implying that identifying the facial expression(s) of affective states is difficult for humans. As the uncertainty interval of the pain–pleasure distance is very close to twice the mode of the pain–pleasure distance (Figure 3), we found a further explanation as to why it is so difficult for humans to visually distinguish between the facial expression of pleasure versus that of pain. Thirdly, the presence of noise contributes to the explanation of why humans have such difficulties when confronted with the task of distinguishing between facial expressions of pain and of pleasure. Fourthly, a clustering algorithm succeeded in identifying patterns in the noise-free pain–pleasure–neutral distances' renditions. Humans are not successful in seeing noise-free distances, thus we cannot expect humans to identify the discovered patterns in these distances.

**Author Contributions:** T.H. collected the videos and chose the frame sequences needed for subsequent analyses. J.B. and D.Ř. designed the research framework. H.P. designed the statistical tests and wrote the computer programs. H.P. and J.B. authored the manuscript. S.B. contributed to the research about human ratings of the affective states. All authors contributed to revisions after critically reading various versions of the manuscript. All authors have read and agreed to the published version of the manuscript.

**Funding:** This study is a result of the research funded by the Czech Science Foundation, namely the project GA ČR 19-12885Y "Behavioral and Psycho-Physiological Response on Ambivalent Visual and Auditory Stimuli Presentation".

**Institutional Review Board Statement:** The study was conducted in accordance with the Declaration of Helsinki, and approved by the Institutional Review Board of Faculty of Science, Charles University, Prague, Czech Republic (protocol code 2018/08 approved on 2 April 2018).

**Informed Consent Statement:** Not applicable.

**Data Availability Statement:** The frames were extracted from commercially available videos. As the videos are proprietary, we can only make the extracted frames we used available (upon request) from the corresponding author.

**Conflicts of Interest:** The authors declare no conflict of interest.

**Appendix A**

We used routines supplied by Wolfram Research. We used MATHEMATICA v12.4 for this study.

In the (explicit) code below, the commands available in the MATHEMATICA package(s) are written in bold. Variables, parameters, etc., that are included by the authors in the code are not.

MATHEMATICA uses its own font, namely 'Consolas', which cannot be imported into this camera-ready document for all lines of the program code. The command lines are therefore written in Calibri and the explanatory text in Palatino Linotype. The explanatory comments are marked by bullet points, followed by a TAB.

- Every user has his/her own folder structure. By "*path*" (below), we mean the path to the folders containing the video frames.
- The command line **Join[ ... ]** is long because it loads a segment of the video dynamically. It is modified accordingly for other videos loaded for further frame extraction.

  **SetDirectory[**"*path*"**]**

  jpgApleasure **= Join[Flatten[Table[StringJoin[**"A_pleasure"**,StringJoin[{ToString [0],  ToString[i]}]**"**.jpg],{i,1,9}],**Flatten[Table[StringJoin[**"A_Pleasure"**,StringJoin[{ToString[j], ToString[i]}],{i,0,9}],{j,1,4}],{StringJoin[**"A_Pleasure"**,StringJoin[{ToString [5], ToString [0]}],**"**.jpg]}]**]

  face **= Import[**jpgApleasure[[25]]**]**

  faceApleasure **= FindFaces[**face,"Image"**,Method**⟶"Haar"**,PaddingSize**⟶30**]**

- The above structure is suitably modified for the other faces of Actress A.

```
faceAneutral = . . . ;
faceAneutral2 = . . . ;
faceAneutral3 = . . . ;
faceApain = . . . ;
```

- The five faces are aligned.

```
{faceAneutral,faceA,faceneutral2,faceApain,faceApleasure,faceAneutral3};
faceAjoin = FaceAlign[%,Automatic,{60,60},PerformanceGoal⟶"Quality"];
```

- The proprietary code from MATHEMATICA uses AI (internally trained) to extract feature vectors from the list of faces.

```
faceAextjoin = FeatureExtraction[faceAjoin];
```

- The proprietary code from MATHEMATICA uses a neural network to dimension-reduce the feature vectors.

```
faceAextractReduce = DimensionReduce[faceAextjoin,2,RandomSeeding⟶Prime [137]];
MeanAneutralReduce = Mean[Table[faceAextractReduce[[i]],{i,{1,2,5}}]];
```

- The (Euclidean) distances are computed.

```
distApain = Norm[faceAextractReduce[[3]]-meanAneutralReduce];
distApleasure = Norm[faceAextractReduce[[4]]-meanAneutralReduce];
distAPainPleasure = Norm[faceAextractReduce[[3]]-faceAextractReduce[[4]]];
neutralDistances = {Table[Norm[faceAextractReduce[[i]]-
Mean[Table[faceAextractReduce[[j]],{j,{1,2,5}}]]]],{i,{1,2,5}]
Table[Norm[faceTextractReduce[[i]]-
Mean[Table[faceTextractReduce[[j]],{j,{1,2,5}}]]]],{i,{1,2,5}]}
```

- The above steps are repeated for the other 19 × 5 faces; A⟶B, A⟶C, . . . and so on up to and including A⟶T.

```
painDistances = {distApain, . . . . . . distTpain}
pleasureDistances = {distApleasure, . . . . . . distTpleasure}
distancesPainPleasure = {distAPainPleasure, . . . , . . . ,distTPainPleasure}
```

- The commands below are used to find the ML distribution of the distances.

```
distributionList = {NormalDistribution[μ,σ], LogNormalDistribution[μ,σ], WeibullDis-
tribution[k,λ], GammaDistribution[k,θ]};
data = distancesPainPleasure;
distNorm = EstimatedDistribution[data,distributionList[[1]]]
LLnorm = LogLikelihood[%,data]
distLogNorm = EstimatedDistribution[data,distributionList[[2]]]
LLlogn = LogLikelihood[%,data]
distWeib = EstimatedDistribution[data,distributionList[[3]]]
LLweib = LogLikelihood[%,data]
distGamm = EstimatedDistribution[data,distributionList[[4]]]
LLgamm = LogLikelihood[%,data]
logLikeList = { LLnorm,LLlogn,LLweib,LLgamm}
posML = Flatten[Position[%,Max[%]]][[1]]
distML = distributionList[[%]]
```

- The code below is used to determine the $\text{HDI}_{95\%}$ uncertainty interval. Note that the precision arithmetic requires several hundred (decimal) digits.

```
distA = %;
weib = distML;
```
$$\text{modeWeib} = \left(1 - \frac{1}{\%[[1]]}\right)^{\frac{1}{\%[[1]]}} \%[[2]]$$
```
modeA = %;
```

$$
\begin{aligned}
\text{solutions3} = \quad &\textbf{Reverse}[\textbf{Table}[\textbf{NSolve}[\{g == \textbf{SetPrecision}[i(g/.u \\
&\rightarrow \text{modeA}), 5\textbf{MachinePrecision}], 0 < u \\
&< 1\}, u, \textbf{Reals}]//\textbf{Quiet}, \{i, 0.004, 0.24, 0.002\}]];
\end{aligned}
$$

$$
\text{s1s2Solutions3} = \textbf{Table}[\{u/.\text{solutions3}[[i, 2]], u/.\text{solutions3}[[i, 1]]\}, \{i, 1, \textbf{Length}[\%]\}];
$$

$$
\begin{aligned}
\text{cdfSolutions3} = \quad &\textbf{Table}[\{\textbf{CDF}[\textbf{SetPrecision}[\text{distA}, 5\textbf{MachinePrecision}], u \\
&/.\text{solutions3}[[i, 2]]] \\
&-\textbf{CDF}[\textbf{SetPrecision}[\text{distA}, 5\textbf{MachinePrecision}], u \\
&/.\text{solutions3}[[i, 1]]]\}, \{i, 1, \textbf{Length}[\%\%]\}];
\end{aligned}
$$

$$
\text{funcPts} = \textbf{Transpose}[\{\text{cdfSolutions3}, \text{s1s2Solutions3}\}];
$$

$$
\text{iFun} = \textbf{Interpolation}[\text{funcPts}, \textbf{InterpolationOrder} \rightarrow 1];
$$

$$
\text{s1s2} = \textbf{Sort}[\textbf{SetPrecision}[\text{iFun}[0.95], 5\textbf{MachinePrecision}]];
$$

- The code below calculates the SVD and the approximation using only the first three singular values.

```
matPre =
Table[Flatten[{neutralDistances[[i]],painDistances[[i]],pleasureDistances[[i]]]},{i,1,20}];
Transpose[matPre];
matRed = Transpose[Table[ 1/Mean[%[[i]]] %[[i]] − 1, {i, 1, 5}]];
{Umat3,Smat3,VTmat3} = SingularValueDecomposition[matRed, 3];
mat3 = Umat3.Smat3.Transpose[VTmat3];
```

- The code below generates a list of colors needed for the graphics.

```
Delete[ColorData[3, "ColorList"][[2; ;]], 2];
Join[%,{Darker[Brown, 0.15]}, {Green}, {Cyan}];
Join[%,{Darker[Yellow, 0.15]}, {Lighter[Orange, 0.2]}, {Pink}];
farbe2 = Join[%,{Darker[LightPurple, 0.1]}, {Darker[LightGreen, 0.35]}];
```

- The code below finds the clusters of the SVD-3 approximated coordinates of the affective state distances.

```
clust = FindClusters[mat3, Method → "Agglomerate", RandomSeeding → Prime[137]];
Table[Length[%[[i]]], {i, 1, Length[%]}]
Length[%%]
Flatten[Table[Position[mat3, clust[[j]][[i]]], {i, 1, Length[clust[[j]]]}]];
clustPos = Table[Union[%[[j]]], {j, 1, Length[clust]}];
```

- A suite of graphics routines (not listed) are used to display the results for the manuscript.

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
