# Peer review of "Determination of “Neutral”–“Pain”, “Neutral”–“Pleasure”, and “Pleasure”–“Pain” Affective State Distances by Using AI Image Analysis of Facial Expressions"

_technologies, doi:10.3390/technologies10040075_

Round 1
Reviewer 1 Report
The article addresses the automated determination of the affective state of a person based on images. The procedure is described in the abstract: first, feature extraction from the images, then dimensionality reduction is performed, followed by clustering is employed. The authors mix the expected benefits of the models with the technical details that are thought to produce these benefits.
Provided that the article details the _implementation of AI_, it is staggering the reference to techniques from artificial intelligence under the umbrella name AI. In this context the “AI” image analysis software is the “FindFaces” from Mathematica and from the article one cannot determine the specifics of the algorithm.
The authors proceed with the dimension-reduced and – are they also clustered? – versions of the data to find the average values – as displayed in Figure 2. Given that the average values are used subsequently, the assumption is that the center of the data is representative for the respective classes. It is questionable whether this representativeness holds or not.
The authors compute the “most likely” distribution family of distances between points and include the normal, the log-normal distributions among others. The finding that the distribution with enforced positive values comes out is not surprising, one should have tried only those type of distributions.
Also, the fact that the non-neutral affective states are more concentrated is not surprising either, given that the “relaxed” class is _any_ of that is not in the affective class, therefore the larger spread.
The title of the article is not clear either: what are the “distances” the authors determine? If there is no standard for the distance, can the results be claimed as objective?
In present form the article cannot be accepted, suggest a clarification of goals, a more technical description of the methods (or the complete avoidance of technicalities), and more emphasis on the application of the “AI flavor” that was highlighted in the article.
Reviewer 2 Report
The work studies machine learning methods in image recognition to distinguish between facial expressions of pain and pleasure. I doing so, this work reports results that are extensions of a previously reported pilot study published in 2021. In the previous work, approaches have been investigated, whether it is possible to distinguish the facial expression of pain from that of pleasure in women. The authors conclude from this (extended) work, that image analysis software can also be used to construct feature vectors of the facial expressions of pain and of pleasure and compared them with the feature vectors of the neutral facial expressions. I suggest to incorporate a native speaker (or a professional proof reading service) for proof reading the manuscript, for example:
“The analyses of the outcomes are rich in information. We found far more than we had been looking for.”
These sentenced may be grammatically correct, but are not appropriate for a scientific publication (in a technical journal).
Reviewer 3 Report
This paper shows how the use of AI technology that analyzes images are superior to humans' abilities to solve these same facial expression decoding challenges. They use image analysis software to extract feature vectors of the facial expressions neutral, pain, and pleasure displayed by 20 actresses. As a result, AI image recognition methods are superior to human abilities in distinguishing between facial expressions of pain and pleasure. Statistical methods and hierarchical clustering offer possible explanations for why humans fail. The reliability of commercial software, which attempts to identify facial expressions of affective states, can be improved by using the results of our analyses.
I believe this research is designed appropriately, and the experimental results support the conclusions. However, this paper requires editing of English and writing style.
- Some paragraphs have only one or two sentences. Add more sentences.
- I think terms like 'to wit' and 'aim' are unnecessary in this paper.
- The caption of figure 1 is too long. Figure 1 is not a particular neural network model.
- BDSM stands for? When you use short terms, describe the full word the first time.
- In line 179, the authors don't need to mention "one co-author (T.H)." Just say, "we scanned~."
Round 2
Reviewer 2 Report
I thank the authors for addressing my comments.
This manuscript is a resubmission of an earlier submission. The following is a list of the peer review reports and author responses from that submission.
Round 1
Reviewer 1 Report
This paper presents a method for detecting human emotion, especially pain versus pleasure, by examining facial images using a neural network and statistical methods.
Is there any way other researchers in this field can verify your results? You provide no details of the image processing software. Will your method work for images from ImageNet? Can you place your input images on a publicly accessible home page for download? If not, then I cannot recommend this paper for publication.
I would like to see more details about the neural network employed. You say there are seven layers, how are they connected? What are the inputs and outputs? Can you provide a diagram of your network? What is it based on? Did you use a library, like Keras or PyTorch? Can other researchers access your Python code?
What is the benefit to society in general of being able to automatically distinguish between pain and pleasure? You mention that the same technology can be applied to breast cancer detection, but if your goal is diagnosis, why not concentrate on medical images? Are there other applications for this technology?
Why is the distinction between pain and pleasure important? It seems ambiguous to me. Some people may get pleasure from pain. It seems impossible to distinguish the two quantitatively.
Can you provide an example of your 20 x 5 matrix?
Is the lengthy discussion about diamond cutters really necessary?
Did you test human evaluation of pleasure and pain on the same images used for the automatic test?

Reviewer 2 Report
Respecting the authors’ work, I would like to propose the following modifications, which can enhance the reliability and validity of their work:
- Research questions, that drive the paper, should be built in the introduction from an ongoing and pertinent bibliography (up to 2021-22) and these should be of global interest and not focused on a particular local problem. Identifying a research gap is the most important by indicating in-text some newer references that are significant to your particular field of research.
- The problem statement needs to be clear into Introduction.
- Some indicative review articles regarding the use of AI technologies that the same authors want to utilize maybe of great importance to readers in order to identify the research “gap” more easily without mention too many previous studies. Additionally, what are the potentials of using such technology?
- The authors should make explicit suggestions about how their study affects the design or use of educational computer systems. Is there something new about a particular theory, or is there evidence of theory advancement?
- The added value of authors’ work is not clear in terms of proper and current research (up to 2021-22).
- Authors should answer your research question in the conclusions and discussion. Please provide a reasonable need to read your work’s results than previous ones or simply answer what we learned compared with current, significant research (up to 2021 should be your work’s “significance”).
- How general are your results and how do you believe that such findings have to be of global interest? Please relate these with your limitations and Discussion that is not exist. Why?
- Practical and educational implications are not provided too.
Reviewer 3 Report
Trying to replace subjective human related and conditioned classification, specially where facial expressions are involved, by automated "objective" mechanisms is an extremely difficult task. Even if results scientifically sound are obtained referring to simplified particular situations, ie with a sample representative and significant enough with a quantified, low, variability and uncertainty, some people can, with total legitimacy, argue against including because for different persons in different cultures and or environments a "sad" looking face may mean a different thing... And these difficulties are evident throughout the text and not fully resolved with this work.
The image processing methods are not presented and explained in sufficient detail: Chapter 2.2. Methods "We use AI image analysis software to:"- a large number of routines exist to perform the kinds of tasks you refer. Which were use, why and how? Thats a fundamental discussion to be made in order to be able to interpret and discuss the results in order to establish sound conclusions.
The conclusions are not set in a clear and objective way for instance:
. "For one, AI methods can more reliably distinguish between pain and pleasure than humans can." What is the meaning of reliability in this context?
How AI is implemented? Is there no human input? If not so, how can we give more value to the conclusion on the effectiveness of AI results than human discrimination of states?
"Fourthly, a clustering algorithm could identify patterns in the noise-free pain-pleasure-neutral distances." What the quality of this statement? What is the meaning of "could" in this statement?